# Prevalence of carbapenem resistance in Acinetobacter baumannii and Pseudomonas aeruginosa in sub-Saharan Africa: A systematic review and meta-analysis

Margaret Toluwalayo Arowolo[1]*, Oluwatosin Qawiyy Orababa[2], Morufat Oluwatosin Olaitan[3], Bisola Victoria Osibeluwo[1], Utibeima Udo Essiet[1], Olajumoke Hanah Batholomew[4], Oluwaseyi Gbotoluwa Ogunrinde[1], Oluwaseyi Aminat Lagoke[1], Jeffrey Difiye Soriwei[5], Olanrewaju David Ishola[6], Onyinye Maryann Ezeani[1], Aminat Oyeronke Onishile[7], Elizabeth Olumodeji[8]

1 Department of Microbiology, University of Lagos, Akoka, Lagos, Nigeria, 2 School of Life Sciences, University of Warwick, Coventry, United Kingdom, 3 Department of Microbiology, University of Ibadan, Ibadan, Nigeria, 4 School of Health Sciences, University of Bristol, Bristol, United Kingdom, 5 Department of Public Health in Microbiology, University of Bedfordshire, Luton, United Kingdom, 6 Department of Biomedical Science, Nottingham Trent University, Nottingham, United Kingdom, 7 Faculty of Health Studies, School of Nursing and Healthcare Leadership, University of Bradford, West Yorkshire, England, United Kingdom, 8 Department of Biology, Texas Women University, Denton, TX, United States of America

☯ These authors contributed equally to this work.
* arowolotoluwalayo@gmail.com

## Abstract

Carbapenems are effective drugs against bacterial pathogens and resistance to them is considered a great public health threat, especially in notorious nosocomial pathogens like *Acinetobacter baumannii* and *Pseudomonas aeruginosa*. In this study, we aimed to determine the prevalence of carbapenem resistance in *A. baumannii* and *P. aeruginosa* infections in Sub-Saharan Africa. Databases (PubMed, Scopus, Web of Science, and African Journal Online) were systematically searched following the Preferred Reporting Items for Systematic review and meta-analysis protocols (PRISMA-P) 2020 statements for articles reporting carbapenem-resistant *Acinetobacter baumannii* (CRAB) and carbapenem-resistant *Pseudomonas aeruginosa* (CRPA) prevalence between 2012 and 2022. Pooled prevalence was determined with the random effect model and funnel plots were used to determine heterogeneity in R. A total of 47 articles were scanned for eligibility, among which 25 (14 for carbapenem-resistant *A. baumannii* and 11 for carbapenem-resistant *P. aeruginosa*) were included in the study after fulfilling the eligibility criteria. The pooled prevalence of CRPA in the present study was estimated at 8% (95% CI; 0.02–0.17; $I^2$ = 98%; P <0.01). There was high heterogeneity (Q = 591.71, $I^2$ = 98.9%; P<0.0001). In addition, this study's pooled prevalence of CRAB was estimated at 20% (95% CI; 0.04–0.43; $I^2$ = 99%; P <0.01). There was high heterogeneity (Q = 1452.57, $I^2$ = 99%; P<0.0001). Also, a funnel plot analysis of the studies showed high degree of heterogeneity. The carbapenemase genes commonly isolated from *A. baumannii* in this study include $bla_{OXA23}$, $bla_{OXA48}$, $bla_{GES}$, $bla_{NDM}$, $bla_{VIM}$, $bla_{OXA24}$, $bla_{OXA58}$, $bla_{OXA51}$, $bla_{SIM-1}$, $bla_{OXA40}$, $bla_{OXA66}$, $bla_{OXA69}$, $bla_{OXA91}$, with

**Data Availability Statement:** All relevant data are within the manuscript and its Supporting Information files.

**Funding:** The authors received no specific funding for this work.

**Competing interests:** The authors have declared that no competing interests exist.

$bla_{OXA23}$ and $bla_{VIM}$ being the most common. On the other hand, $bla_{NDM}$, $bla_{VIM}$, $bla_{IMP}$, $bla_{OXA48}$, $bla_{OXA51}$, $bla_{SIM-1}$, $bla_{OXA181}$, $bla_{KPC}$, $bla_{OXA23}$, $bla_{OXA50}$ were the commonly isolated carbapenemase genes in *P. aeruginosa*, among which $bla_{VIM}$ and $bla_{NDM}$ genes were the most frequently isolated. Surveillance of drug-resistant pathogens in Sub-Saharan Africa is essential in reducing the region's disease burden. This study has shown that the region has significantly high multidrug-resistant pathogen prevalence. This is a wake-up call for policymakers to put in place measures to reduce the spread of these critical priority pathogens.

## Introduction

Antimicrobial resistance (AMR) is a leading public health threat globally, considerably escalating morbidity, mortality and treatment failure of microbial infections, as well as economic losses to individuals and nations [1]. According to the 2016 Review on Antimicrobial Resistance, AMR would be responsible for 10 million deaths yearly by 2050 with a large amount of these deaths occurring in Sub-Saharan Africa [2]. However, a recent report of almost 5 million deaths linked with AMR in 2019 alone has shown that we will be reaching the 10 million AMR-associated deaths sooner than earlier predicted [3].

Carbapenems are beta-lactam antibiotics with broad-spectrum bactericidal activities against both Gram-positive and negative pathogens [4]. They are often used as last-line drugs against bacterial infections [5]. Unfortunately, bacterial pathogens have developed resistance to this last-resort group of antibiotics through various genetic modifications and production of carbapenem-hydrolysing enzymes [6]. Treatment of infections have become more difficult and expensive, especially against the notorious Gram-negative bacteria, due to resistance to last-line antibiotics.

*Acinetobacter baumannii* and *Pseudomonas aeruginosa* are Gram-negative pathogens that belong to the ESKAPE group (an acronym for the group of bacteria, encompassing both Gram-positive and Gram-negative species, made up of *Enterococcus faecium*, *Staphylococcus aureus*, *Klebsiella pneumonia*, *Acinetobacter baumannii*, *Pseudomonas aeruginosa* and *Enterobacter species* [7]. These are bacterial pathogens that are notorious for their resistance to clinically relevant antimicrobials. Furthermore, carbapenem-resistant *A. baumannii* (CRAB) and *P. aeruginosa* (CRPA) have both been grouped as critical priority pathogens for which there is a need to develop new and effective antimicrobials. These have made the surveillance of these two pathogens a necessity, especially in low-middle-income countries where surveillance is poor. A recent report shows that CRAB and CRPA have mortality rates of 30.5% and 24.5% within 90 days of a positive culture [8]. Moreover, CRAB and CRPA were reported to be responsible for 57,700 and 38,100 deaths globally in 2019 respectively [3].

Sub-Saharan Africa (SSA) is known for having a high burden of infectious diseases which might be linked to the poverty level and poor water, sanitation, and hygiene (WASH) practices in the region [9,10]. The [3] report also showed that Sub-Saharan Africa suffers the highest prevalence of AMR-associated death globally. The scarcity of AMR data from SSA has made it difficult to determine the true risk and burden of AMR infections in the region. Also, a recent study predicting the global prevalence of carbapenemase-producing *P. aeruginosa* could not include majority of Sub-Saharan African countries in the analysis due to scarcity of data. This further emphasizes the need to surveillance of AMR pathogens in Africa, especially in Sub-Saharan Africa. To the best of our knowledge, there is currently no up-to-date systematic

review and meta-analysis that reports the pooled prevalence of carbapenem-resistant *P. aeruginosa* and *A. baumannii* in Sub-Saharan Africa. This systematic review is an extensive analysis of the prevalence of carbapenem-resistant *A. baumannii* with the aim to describe the epidemiology within Sub-Saharan Africa. This study also investigated the prevalence of carbapenemase genes in the region. This would provide insight on the public health risks posed by these priority pathogens and the development of sustainable prevention and control interventions in this region.

## Method

### Search strategy

This study was performed according to the Preferred Reporting Items for Systematic Review and Meta-analysis (PRISMA) guidelines [11]. Electronic databases (PubMed, Scopus, Web of Science, and African Journal Online) were searched for articles published on carbapenem-resistant *A. baumannii* and/or *P. aeruginosa* in sub-Saharan Africa. The last search was on the 31st of July 2022.

### Eligibility criteria

This study only included studies published in Sub-Saharan Africa between January 2012 and July 2022. Only studies that reported *A. baumannii* and/or *P. aeruginosa* prevalence in humans were considered for the analysis. Studies that reported *A. baumannii* and/or *P. aeruginosa* prevalence from countries outside Sub-Saharan Africa were excluded from the analysis. Also, only studies that reported at least one case of carbapenem resistance in *A. baumannii* and/or *P. aeruginosa* were included. Only studies published in English or with English translations available were included in the final analysis. Lastly, cross-sectional studies, retrospective, and longitudinal analyses were considered in this analysis. Literature and systematic reviews were excluded from this analysis.

### Screening strategy

Articles were first screened based on their titles and abstracts by two independent researchers. Two other researchers then screened eligible articles by reading through their full text, ineligible articles were excluded and the reasons for their exclusion were stated (Fig 1). All disagreements were resolved through discussions. A third researcher confirmed the eligibility of the included studies before including them in the analysis. During extraction, for studies that reported different percentages for imipenem or meropenem, preference was given to meropenem since it has better activity against Gram-negative bacteria [12].

### Data extraction and quality appraisal

A data extraction table was created by one reviewer and two other reviewers extracted the following data from the eligible articles; first author, study design, study country, study aim, sample size, sample source, organism prevalence, carbapenem-resistance prevalence, Antimicrobial Susceptibility Testing (AST) method, carbapenem used. All disagreements were resolved through discussions and a fourth reviewer confirmed the extracted data. This step was performed for each of the organisms.

### Data analysis

The data extracted was cleaned for any eligibility criterion error, and the meta-analysis was performed using RStudio version 4.2.0. The Carbapenem-resistant *A. baumannii* (CRAB) and

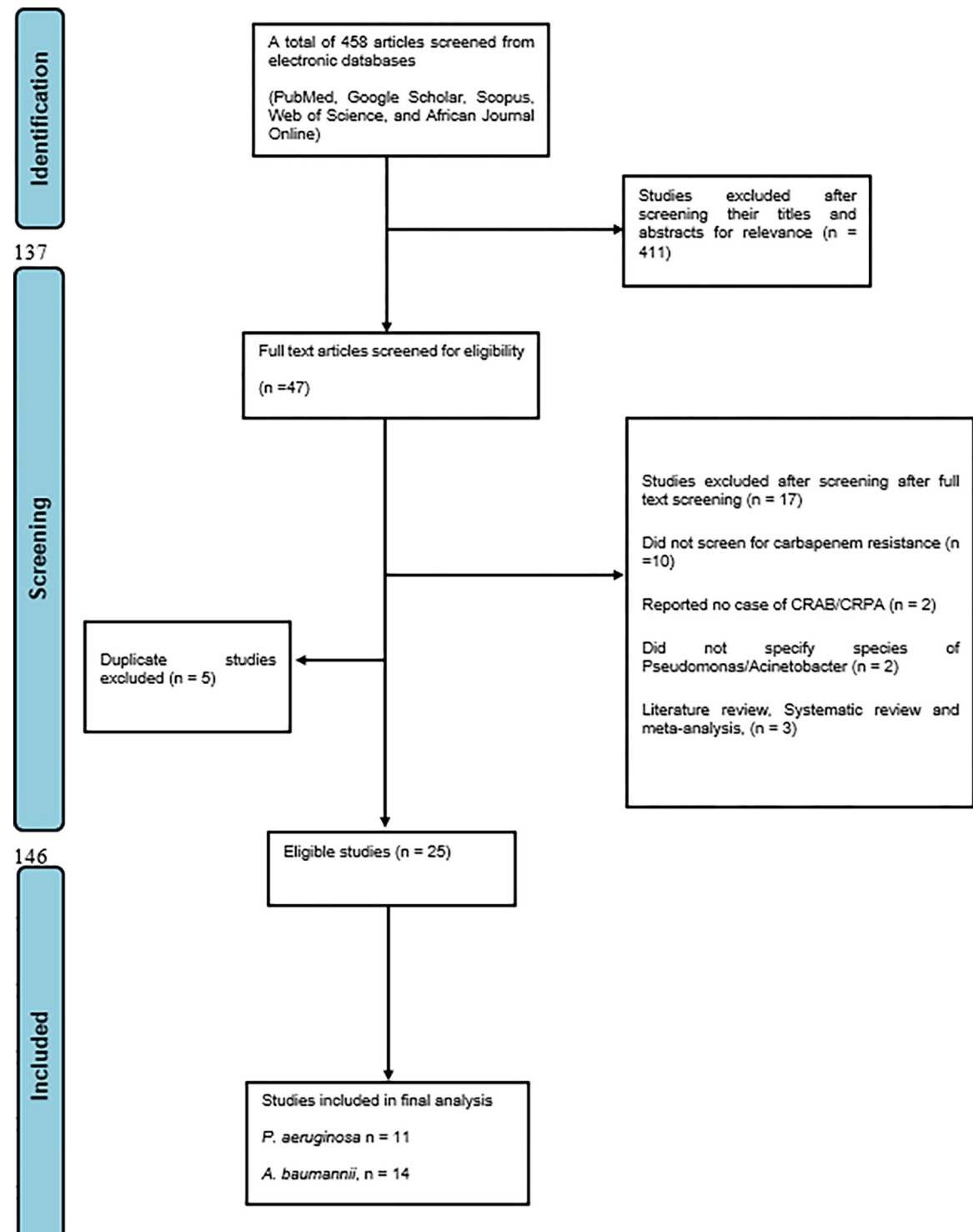

**Fig 1. PRISMA flowchart for the search process.**

*P. aeruginosa* (CRPA) pooled prevalence of the present study was analyzed using the random-effect model while subgroup prevalence was considered based on the source of the reported studies. Cochran Q statistics and the $I^2$ (inverse variance index) were used to analyze the

heterogeneity (low, 0–0.25; fair, 0.25–0.5; moderate, 0.5–0.75; and high, above 0.75). A p-value of <0.01 was considered significant. Publication bias and degree of heterogeneity was examined visually with the use of the funnel plot.

## Result

A total of 25 articles (CRAB, 14; CRPA 11) (Fig 2) were included in the final analysis after screening 458 articles from different electronic databases. These articles were from different countries and sub-regions of sub-Saharan Africa. Of the 14 articles analysed for the CRAB, South Africa had 4(28.6%), Sudan and Uganda had 2(14.3%) each, while Ethiopia, Senegal Malawi, Kenya, Nigeria, and Sierra Leone had one each. For the CRPA analysis, Nigeria, Ethiopia, Uganda, and Sudan each had 2 eligible articles included in the final analysis while Malawi, Ghana, and Kenya each had just one eligible article (Tables 1 and 2).

### Meta-analysis of Carbapenem-resistant *A. baumannii*

The pooled prevalence of CRAB in the present study was estimated at 20% (95% Confidence Interval [CI]; 0.04–0.43; $I^2$ = 99%; P <0.01) (Fig 3). The funnel plot (Fig 4) and Q statistics show high heterogeneity (Q = 1452.57, $I^2$ = 99%; P<0.0001) (S1 Fig) between the CRAB studies.

### Meta-analysis of carbapenem-resistant *P. aeruginosa*

The pooled prevalence of CRPA in the present study was estimated at 8% (95% CI; 0.02–0.17; $I^2$ = 98%; P <0.01) (Fig 5). Similarly, there was high heterogeneity between CRPA studies analysed as shown by the Q statistics (Q = 591.71, $I^2$ = 98.9%; P<0.0001) and the funnel plot (Figs 6 and S2).

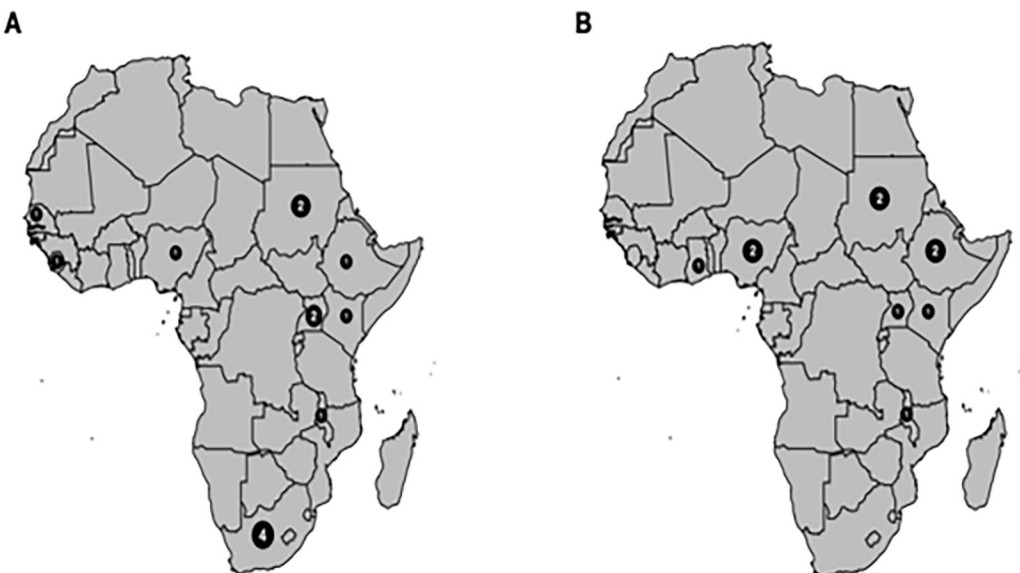

**Fig 2. Distribution of eligible articles included in the final analysis.** A. A total of 14 articles were analysed to determine the pooled prevalence of carbapenem-resistant *Acinetobacter baumannii*. A majority (4) were from South Africa, Sudan and Uganda had 2 each, while Ethiopia, Senegal, Malawi, Kenya, Nigeria, and Sierra Leone had one each. B. A total of 11 articles were analysed to determine the pooled prevalence of carbapenem-resistant *Pseudomonas aeruginosa*, 2 each from Ethiopia, Nigeria, and Sudan while Malawi, Ghana, Kenya, and Uganda had one each.

**Table 1. Data extraction table for carbapenem-resistant _A. baumannii_ (CRAB) studies.**

| First author | Year | Study design | Study country | Study aim | Sample size | Sample source | _A. baumannii_ prevalence | Carbapenem-resistant _A. baumannii_ prevalence (%) | AST method | Carbapenem use |
|---|---|---|---|---|---|---|---|---|---|---|
| [13] | 2021 | cross-sectional study | Sudan | To assess the phenotypic and genotypic patterns of antimicrobial resistant strains of _Acinetobacter baumannii_ at hospital settings, Khartoum, Sudan | 36 | Human | 36 | 17 (47%) | Disc diffusion | Imipenem |
| [14] | 2013 | cross-sectional | South Africa | To evaluate and optimise multiplex polymerase chain reaction (M-PCR) assays for the rapid differentiation of the four subgroups of the OXA genes and the subgroups of the MBL genes of _A. baumannii_. | 97 | Human | 97 | 61 (62.8%) | Vitek 2 | Meropenem and Imipenem |
| [15] | 2021 | Longitudinal | Ethiopia | To detect and phenotypically characterize carbapenem-resistant Gram negative bacilli from Ethiopian public health institute | 1337 | Human | 36 | 4 (0.3%) | Simplified carbapenem inactivation method | Meropenem |
| [16] | 2016 | cross sectional study | Uganda | To determine the prevalence of carbapenem-resistant _P. aeruginosa_ and _A. baumannii_ at Mulago Hospital in Kampala Uganda, and to establish whether the hospital environment harbours carbapenem-resistant Gram-negative rods | 869 | Human | 29 | 9 (1.04%) | Disc diffusion | Imipenem |
| [17] | 2015 | cross-sectional study | South Africa | To determine the prevalence of β-lactamase genes in multidrug-resistant (MDR) clinical _A. baumannii_ isolates using Multiplex-PCR (M-PCR) assays. | 94 | Human | 94 | 80 (85.1%) | Vitek 2 | Meropenem and Imipenem |
| [18] | 2019 | cross sectional study | South Africa | To investigated the 28 genetic determinants of multi-drug resistant _A. baumannii_ (MDRAB) at a teaching hospital in 29 Pretoria, South Africa. | 100 | Human | 100 | 95 (95%) | Vitek 2 | Imipenem and Meropenem |
| [19] | 2012 | longitudinal prospective study | Senegal | To identify the presence of _A. baumannii_ carbapenem-resistant encoding genes in natural reservoirs in Senegal, where antibiotic pressure is believed to be low | 717 | Human | 78 | 6 (0.8%) | Disc diffusion | Imipenem |
| [20] | 2019 | cross-sectional | Sudan | To characterize the genotypes and phenotypes associated with carbapenem-resistance in Gram-negative bacilli from patients in Sudan | 367 | Human | not stated | 1 (0.3%) | Disc diffusion | Meropenem |
| [21] | 2022 | cross-sectional study | Malawi | To ascertain the antimicrobial resistance in clinical bacterial pathogens from in-patient adults in a tertiary hospital in Malawi | 694 | Human | 26 | 4 (0.58%) | Disc diffusion | Meropenem |
| [22] | 2021 | cross sectional study | Kenya | To understand the antibiotic resistance profiles, genes, sequence types, and distribution of carbapenem-resistant gram from patients in six hospitals across five Kenyan counties. | 48 | Human | 27 | 27 (56%) | Disc diffusion | Meropenem |
| [23] | 2013 | cross sectional study | Nigeria | To report the presence of carbapenem-encoding genes in imipenem-resistant _A. baumannii_ among multidrug-resistant clinical isolates collected from the University College Hospital, Ibadan, south-western Nigeria. | 5 | Human | 3 | 3 (60%) | Combined disc diffusion test (CDDT) | Imipenem |

_(Continued)_

**Table 1.** (Continued)

| First author | Year | Study design | Study country | Study aim | Sample size | Sample source | *A. baumannii* prevalence | Carbapenem-resistant *A. baumannii* prevalence (%) | AST method | Carbapenem use |
|---|---|---|---|---|---|---|---|---|---|---|
| [24] | 2017 | cross-sectional | Uganda | To determine the intra-species genotypic diversity among *P. aeruginosa* and *Acinetobacter baumannii* isolated from hospitalized patients and the environment at Mulago Hospital | 736 | Human | 7 | 1 (0.14%) | Disc diffusion | Meropenem |
| [25] | 2020 | cross-sectional | Sierra Leone | To assess antibiotic resistance rates from isolates in the urine and sputum samples of patients with clinical features of healthcare-associated infections HAIs. | 164 | Human | 16 | 2(1.2%) | Vitek 2 | Meropenem and Imipenem |
| [26] | 2018 | retrospective descriptive study | South Africa | To determine prevalence of culture confirmed sepsis due to *A. baumannii*, antimicrobial susceptibility and case fatality rates (CFR) due to this organism | 93527 | Human | 399 | 11 (0.01%) | Disc diffusion | Meropenem and Imipenem |

CRAB = Carbapenem-resistant *Acinetobacter baumannii*.

AST = Antimicrobial Susceptibility Testing.

*The percentage in the bracket is a fraction of the sample size.

## Prevalence of carbapenem-resistant genes in *A. baumannii* and *P. aeruginosa* in Sub-Saharan Africa

The carbapenemase genes isolated from *A. baumannii* reported in the articles analysed include $bla_{OXA23}$, $bla_{OXA48}$, $bla_{GES}$, $bla_{NDM}$, $bla_{VIM}$, $bla_{OXA24}$, $bla_{OXA58}$, $bla_{OXA51}$, $bla_{SIM-1}$, $bla_{OXA40}$, $bla_{OXA66}$, $bla_{OXA69}$, $bla_{OXA91}$. The most common carbapenemase gene in the studies analysed are the $bla_{OXA23}$ and $bla_{VIM}$. On the other hand, the carbapenemase genes reported in *P. aeruginosa* from studies in Sub-Saharan Africa included in this analysis are $bla_{NDM}$, $bla_{VIM}$, $bla_{IMP}$, $bla_{OXA48}$, $bla_{OXA51}$, $bla_{SIM-1}$, $bla_{OXA181}$, $bla_{KPC}$, $bla_{OXA23}$, $bla_{OXA50}$. The most frequent among them are the $bla_{VIM}$ and $bla_{NDM}$ genes.

## Discussion

Carbapenems are important broad-spectrum antimicrobials of last-resort; hence, resistance to them signifies increased infection mortality, hospital stay duration, and cost of treatment [31]. This challenge is mostly common to infections associated with notorious clinical pathogens such as *A. baumannii* and *P. aeruginosa* [32]. These two Gram-negative pathogens have been linked to varieties of hospital-acquired infections and multidrug resistance.

Carbapenemase-resistant *A. baumannii* in Sub-Saharan Africa in this study was estimated at 20% (95% CI; 0.04–0.43; $I^2$ = 99%; P <0.01). [33] in their study on the occurrence and frequency of hospital-acquired (carbapenem-resistant) *A. baumannii* in Europe (EUR), Eastern Mediterranean (EMR) and Africa (AFR) stated similar results with a pooled incidence of Hospital Acquired-CRAB of 21.4 (95% CI 11.0–41.3) cases per 1,000 patients in the EUR, EMR and AFR WHO regions. On the other hand, our study revealed the pooled prevalence of CRPA in Sub-Saharan Africa to be 8% (95% CI; 0.02–0.17; $I^2$ = 98%; P <0.01). This value is relatively low compared to the prevalence of carbapenem resistance reported in Indonesia, India, Italy, China, Germany, and Spain [34]. An additional study from Asia recounted a

**Table 2. Data extraction table for carbapenem-resistant *P. aeruginosa* (CRPA) studies.**

| First author | Year | Study design | Study country | Study aim | Sample size | Sample source | *P. aeruginosa* prevalence | Carbapenem-resistant *P. aeruginosa* prevalence (%) | AST method | Carbapenem used |
|---|---|---|---|---|---|---|---|---|---|---|
| [21] | 2022 | cross sectional | Malawi | To ascertain antimicrobial resistance (AMR) in clinical bacterial pathogens from in-hospital adult patients at a tertiary hospital in Malawi | 694 | Human | 29 | 18 (2.5%) | Disc diffusion | Meropenem |
| [15] | 2021 | cross sectional | Ethiopia | To detect and phenotypically characterise carbapenemase no-susceptible Gram-negative bacilli at the Ethopian Public health institute | 1337 | Human | 36 | 4(0.2%) | Disc diffusion | Meropenem |
| [16] | 2016 | cross sectional | Uganda | To determine the prevalence of carbapenem-resistant *P. aeruginosa* and *A. baumanni*i at Mulago Hospital in Kampala Uganda, and to establish whether the hospital environment harbours carbapenem-resistant Gram-negative rods | 869 | Human | 42 | 9 (1.15%) | Disc diffusion | Imipenem |
| [27] | 2019 | cross sectional | Nigeria | To investigate the occurrence of MBL-producing bacteria in a healthcare facility in Nigeria. | 110 | Human | Not reported | 13(11.8%) | Combined disc method | |
| [20] | 2019 | cross sectional | Sudan | To characterise the phenotypes and genotypes associated with carbapenems-resistant in gram-negative bacilli isolated | 367 | Human | Not reported | 14(3.8%) | Disc diffusion | Meropenem |
| [28] | 2019 | cross sectional | Ghana | To apply phenotypic and genotypic methods to identify and characterise carbapenem-resistant (CR) Gram-negative bacteria from the hospital environment in Ghana. | 111 | Human | Not reported | 51(45.9%) | Disc synergy | Edoripenem (75%), imipenem (66.7%) and meropenem (58%) |
| [22] | 2021 | Cross-sectional | Kenya | To understand the antibiotic resistance profiles, genes, sequence types, and distribution of carbapenem-resistant Gram negative bacteria from patients in six hospitals across five Kenyan counties by bacterial culture, antibiotic susceptibility testing and whole genome sequence analysis. Culture, antibiotic susceptibility testing, and whole-genome sequence analysis. | 48 | Human | 14 | 14(29.2%) | | Meropenem |
| [29] | 2021 | cross sectional | Nigeria | To determine the incidence and Molecular Characterization of Carbapenemase Genes in Association with Multidrug-Resistant Clinical Isolates of *Pseudomonas aeruginosa* from Tertiary Healthcare Facilities in Southwest Nigeria | 430 | Human | 430 | 71(16.5%) | Disc diffusion | Meropenem and doripenem |
| [24] | 2017 | cross sectional | Uganda | To report the intra-species genotypic diversity among *P. aeruginosa* and *A. baumannii* isolated from hospitalized patients and the environment at Mulago Hospital, using repetitive elements-based PCR (Rep-PCR) genotyping. | 736 | Human | 9 | 3(0.4%) | | Meropenem |

(*Continued*)

**Table 2.** (Continued)

| First author | Year | Study design | Study country | Study aim | Sample size | Sample source | *P. aeruginosa* prevalence | Carbapenem-resistant *P. aeruginosa* prevalence (%) | AST method | Carbapenem used |
|---|---|---|---|---|---|---|---|---|---|---|
| [29] | 2018 | Cross sectional | Sudan | To detect (blaVIM, blaIMP and blaNDM) Metallo-β-lactamase genes in Khartoum state | 200 | Human | 54 | 33(16.5%) | Disc diffusion | Meropenem and/or imipenem. |
| [30] | 2019 | Cross sectional | Ethopia | To identify and determine multi-drug resistant, extended spectrum β-lactamase and carbapenemase producing bacterial isolates among blood culture specimens from pediatric patients less than five years of age from Tikur Anbessa Specialized Hospital using an automated BacT/Alert instrument. | 340 | Human | Not reported | 4(1.1%) | Disc diffusion | Meropenem |

CRPA = Carbapenem-resistant *Pseudomonas aeruginosa*.

AST = Antimicrobial Susceptibility Testing.

*The percentage in the bracket is a fraction of the sample size.

prevalence of 18.9% CRPA in the Asia-Pacific region. The unexpectedly lower prevalence reported in this study might be due to the absence data from many of the Sub-Saharan African countries. The high heterogeneity observed in this study is most likely not due to publication bias but a result of the different prevalence rates reported in the different studies analysed from different parts of Sub-Saharan Africa and the low number of eligible articles analysed in this study (Figs 5 and 6).

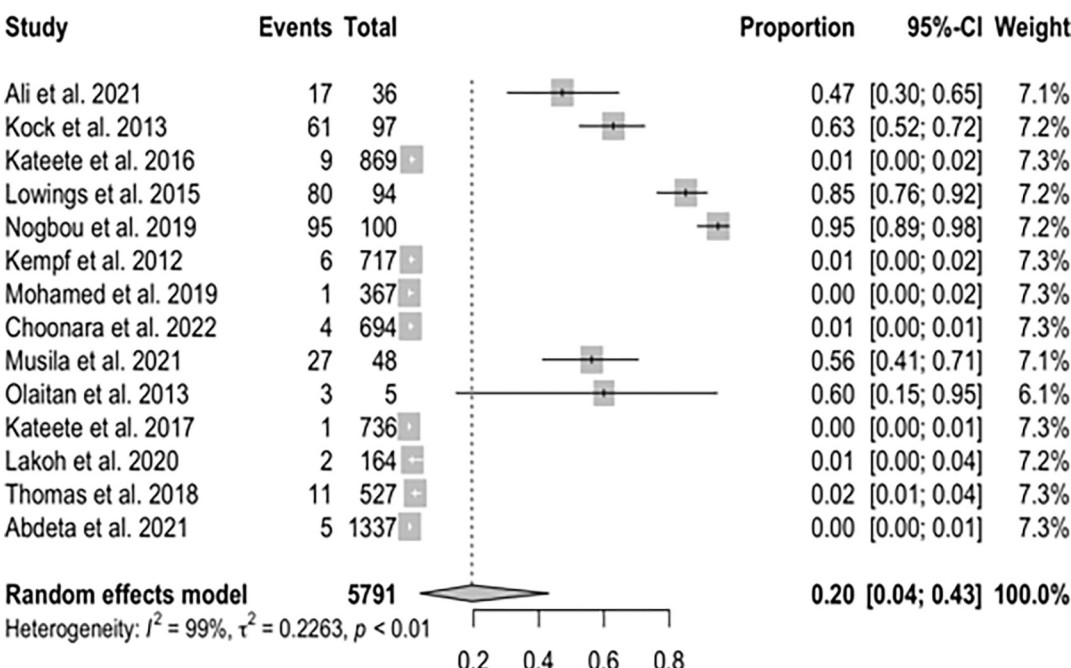

**Fig 3. The Forest plots of random-effects meta-analysis show the pooled prevalence of carbapenem-resistant *Acinetobacter baumannii*.** CI = Confidence interval.

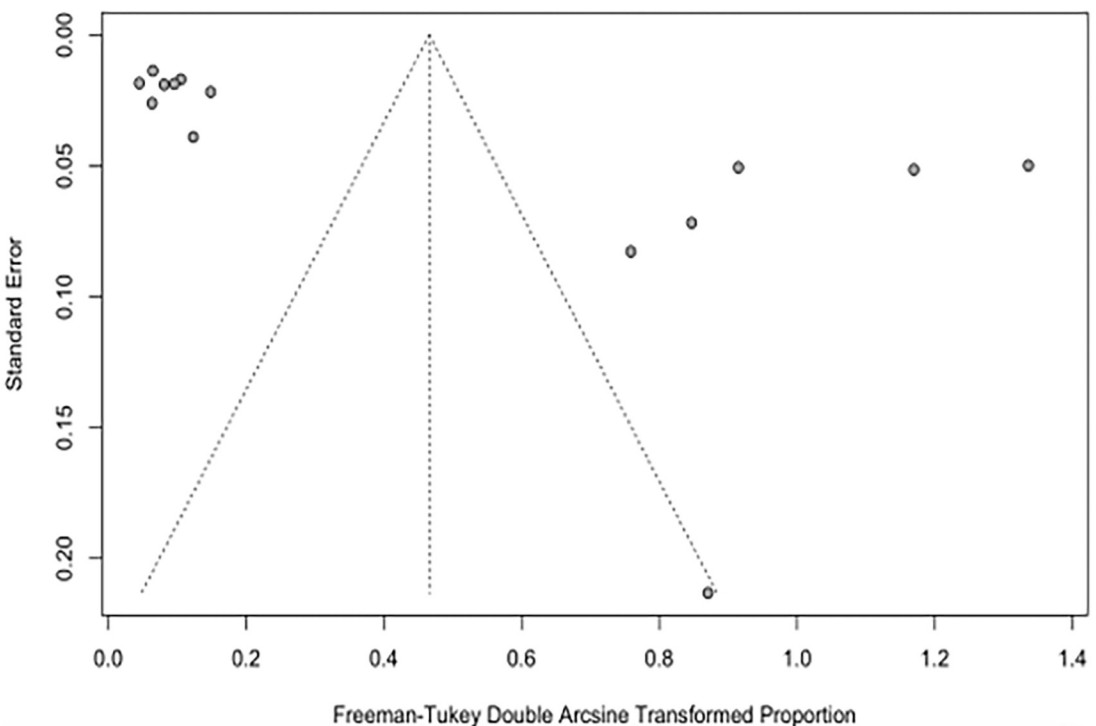

**Fig 4. The Funnel plot of CRAB studies shows the publication bias of the study sample.**

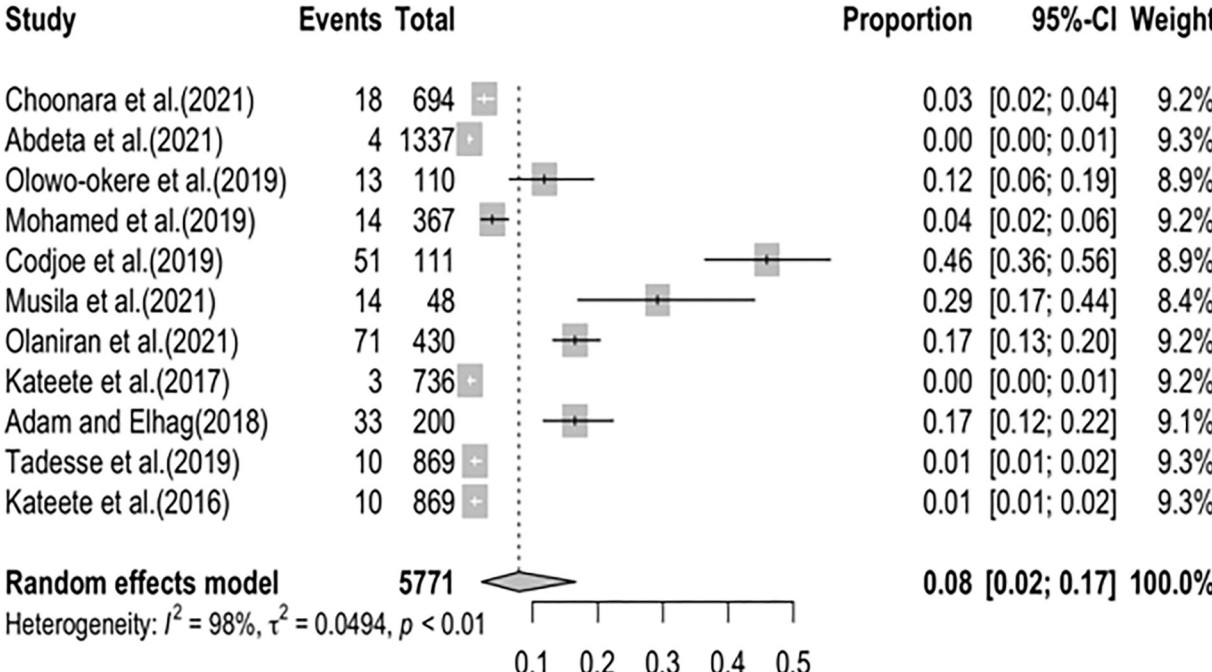

**Fig 5. Forest plot showing the pooled prevalence of carbapenem-resistant *Pseudomonas aeruginosa* in Sub-Saharan Africa.**

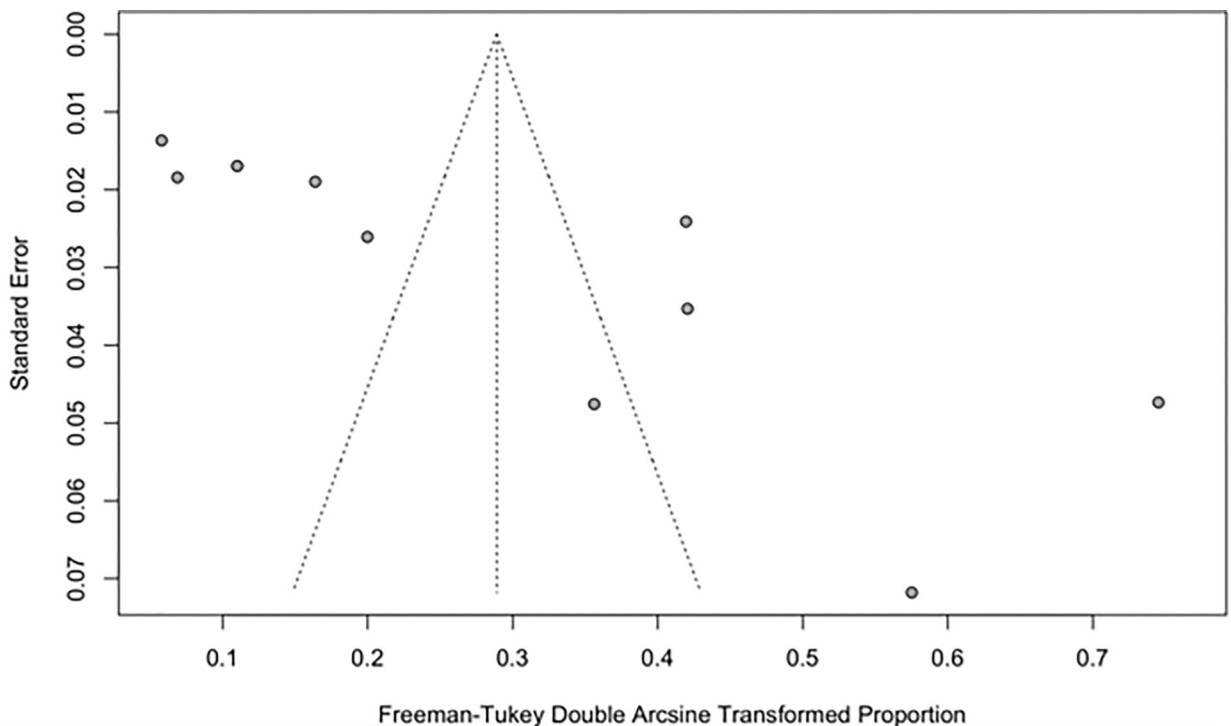

**Fig 6. The Funnel plot of CRPA studies shows the publication bias of the study sample.**

The most common carbapenem-resistant genes in *P. aeruginosa* in the compared studies in this analysis are the NDM and VIM genes. In the reports of [34] the $bla_{VIM}$ and $bla_{IMP}$ were the most prevalent carbapenemase genes in *P. aeruginosa* in Italy and Indonesia. The dominance of OXA-23 genes in CRAB isolates in many of the studies is not surprising as this gene has been associated with carbapenem resistance in this organism since the 1980s [35]. Furthermore, an earlier study indicated that the existence of plasmid encoding OXA-23 alone in *A. baumannii* is sufficient to confer resistance to carbapenem in *A. baumannii* [36]. Even though some of the studies analysed reported carbapenemase genes, many of them did not report the molecular basis of carbapenem resistance in their study. This might be a result of low resources for molecular techniques in many Sub-Saharan African labs.

The pooled prevalence during 2012–2022 should maximally reflect the current status of antibiotic resistance (Tables 1 and 2). Thus, we believe that the current prevalence of antibiotic resistance in *A. baumannii* infection is similar in the Sub-Saharan countries where these studies were carried out, most of which are facing the same degree of severity of antibiotic resistance. Most published studies have concentrated on the hospital epidemiology of these organisms and animal healthcare settings making it difficult to demonstrate the extra-hospital origin of *A. baumannii*. Thus, although antibiotic resistance has long been considered as a modern phenomenon, it predates the concept of selective antibiotic pressure due to clinical antibiotic usage [37].

Over the last few decades, the intensive use of antibiotics in humans and as growth-promoting and as prophylactic agents in livestock have resulted in serious environmental and public health problems since this enhances antimicrobial selective pressure [38]. According to a study by [39], it was discovered that A. *baumannii* isolated from various environmental locations has been linked to nosocomial spread. The resistant bacteria from the extra-hospital

environments may be transmitted to humans, to whom they cause diseases that are difficult to treat with conventional antibiotics.

A major strength of our study is that we included studies comprising both in-patients and out-patients and from a range of different samples, which ensured the representativeness of our estimates for these institutions. However, our study has some limitations. Firstly, the country representativeness of the individual studies is unclear in most cases, limiting our findings' external validity. Secondly, studies are not evenly distributed across the Sub-Saharan regions. Some countries have more studies included in the analysis while there were no studies from others. Consequently, possible differences in CRAB and CRPA incidence and prevalence between the countries may be masked by geographical proximity. Thirdly, due to the relatively low number of hospital-wide studies, our hospital-wide estimates of hospital-acquired *A. baumannii* infections are unlikely to be generalizable.

## Conclusion

*Acinetobacter baumannii* and *Pseudomonas aeruginosa* are two important public health pathogens due to their increasingly multidrug-resistant nature. The prevalence of CRAB and CRPA reported in Sub-Saharan Africa indicates a serious threat to the Sub-Saharan African populace. Sensitization on the dangers of self-medication and poor antibiotics stewardship should be intensified as well as advocacy for the reduction in the use of antibiotics for growth promotion and prophylactics in animals. There is a need for urgent and comprehensive surveillance studies including both hospital environments and communities to determine the true prevalence of these drug-resistant pathogens in sub-Saharan Africa. Moreover, finally, infection control policies promoting personal and environmental hygiene, and appropriate administration of antibiotics by clinicians and veterinarians should be made and enforced to facilitate the reduction of CRAB in the region.

## Supporting information

**S1 Fig. Analysis summary for CRAB.**
(PDF)

**S2 Fig. Analysis summary for CRPA.**
(PDF)

**S1 File.**
(DOCX)

## Author Contributions

**Conceptualization:** Oluwatosin Qawiyy Orababa.

**Data curation:** Margaret Toluwalayo Arowolo, Oluwatosin Qawiyy Orababa, Bisola Victoria Osibeluwo, Olajumoke Hanah Batholomew, Oluwaseyi Aminat Lagoke, Onyinye Maryann Ezeani, Aminat Oyeronke Onishile, Elizabeth Olumodeji.

**Formal analysis:** Oluwaseyi Gbotoluwa Ogunrinde.

**Investigation:** Margaret Toluwalayo Arowolo, Morufat Oluwatosin Olaitan, Jeffrey Difiye Soriwei, Olanrewaju David Ishola, Onyinye Maryann Ezeani, Aminat Oyeronke Onishile.

**Methodology:** Margaret Toluwalayo Arowolo, Oluwatosin Qawiyy Orababa, Morufat Oluwatosin Olaitan, Bisola Victoria Osibeluwo, Utibeima Udo Essiet, Olajumoke Hanah Batholomew, Jeffrey Difiye Soriwei, Olanrewaju David Ishola, Elizabeth Olumodeji.

**Resources:** Margaret Toluwalayo Arowolo, Oluwaseyi Gbotoluwa Ogunrinde.

**Software:** Margaret Toluwalayo Arowolo, Oluwatosin Qawiyy Orababa.

**Validation:** Oluwatosin Qawiyy Orababa.

**Visualization:** Margaret Toluwalayo Arowolo.

**Writing – original draft:** Oluwatosin Qawiyy Orababa.

**Writing – review & editing:** Margaret Toluwalayo Arowolo, Oluwatosin Qawiyy Orababa.

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
