## [Decision Letter · Decision Letter 0]

12 Dec 2022

PONE-D-22-27050

Prevalence of carbapenem resistance in Acinetobacter baumannii and Pseudomonas aeruginosa in sub-Saharan Africa: a systematic review and meta-analysis

PLOS ONE

Dear Dr. Arowolo,

Thank you for submitting your manuscript to PLOS ONE. After careful consideration, we feel that it has merit but does not fully meet PLOS ONE’s publication criteria as it currently stands. Therefore, we invite you to submit a revised version of the manuscript that addresses the points raised during the review process.

For your convenience and attention,  the minor changes that are requested by the Reviewers are available in the Reviewers' reports below. 

We look forward to receiving your revised manuscript.

Kind regards,

Adriano Gianmaria Duse, MD

Academic Editor

PLOS ONE

Journal Requirements:

- https://www.tandfonline.com/doi/full/10.1080/22221751.2019.1698273

-https://www.thefreelibrary.com/Prevalence+and+molecular+analysis+of+multidrug-resistant...-a0618470261

In your revision ensure you cite all your sources (including your own works), and quote or rephrase any duplicated text outside the methods section. Further consideration is dependent on these concerns being addressed

4. Please ensure that you refer to Figures 5, 6 and 7 in your text as, if accepted, production will need this reference to link the reader to the figure.

5. We note that Figure 2 in your submission contain map images which may be copyrighted. All PLOS content is published under the Creative Commons Attribution License (CC BY 4.0), which means that the manuscript, images, and Supporting Information files will be freely available online, and any third party is permitted to access, download, copy, distribute, and use these materials in any way, even commercially, with proper attribution. For these reasons, we cannot publish previously copyrighted maps or satellite images created using proprietary data, such as Google software (Google Maps, Street View, and Earth). For more information, see our copyright guidelines: http://journals.plos.org/plosone/s/licenses-and-copyright.

6. We note you have included a table to which you do not refer in the text of your manuscript. Please ensure that you refer to Table 1 and 2 in your text; if accepted, production will need this reference to link the reader to the Table.

Reviewers' comments:

Reviewer's Responses to Questions

**Comments to the Author**

1. Is the manuscript technically sound, and do the data support the conclusions?

Reviewer #1: Yes

Reviewer #2: Yes

2. Has the statistical analysis been performed appropriately and rigorously? 

Reviewer #1: Yes

Reviewer #2: Yes

3. Have the authors made all data underlying the findings in their manuscript fully available?

Reviewer #1: Yes

Reviewer #2: Yes

4. Is the manuscript presented in an intelligible fashion and written in standard English?

Reviewer #1: No

Reviewer #2: Yes

5. Review Comments to the Author

Reviewer #1: General comments:

1. Absence of page numbers and line numbers.

2. Genus and species are not consistently italicised in the main text.

3. Figures should not be included in the text according to journal recommendations.

4. Spelling errors : ''Acinetobacter baumanii'' for Acinetobacter baumannii'', Discussion - ''tecnics'' for ''techniques'',

table 1 and 2: ''disc'' instead of ''disk''.

5. Word in full prior to using abbreviations: e.g. AST for antimicrobial susceptibility testing.

Abstract:

1. Extra comma after blaimp under results section.

2. Add the funnel plot analysis to the methods.

3. This sentence is not mentioned/explained in the main text ''Carbapenem-resistant A. baumannii prevalence based on

sample source gave estimates of 24% (95% CI; 6 – 49; I2=99%; P<0.01). ''

Introduction:

1. It would be beneficial to mention the global statistics with regards to the prevalence of carbapenem resistant

Acinetobacter baumannii and Pseudomonas aeruginosa here.

2. Prevalence of the common carbapenemase genes worldwide amongst CRAB and CRPA should be highlighted here.

3. Please reword this sentence: ''This has made infection therapy more difficult and expensive, especially against the

notorious Gram-negative bacteria''

4. Define ''ESKAPE group.''

Methods:

1. Need to define the study period accurately and include the months in 2012 and 2022 as the start and end dates.

2. Were all studies that meet the eligibility criteria, irrespective of language, included?

Results:

1. For the carbapenem resistant Pseudomonas aeruginosa (CRPA), a total of 11 articles reviewed, however, only 9 articles

mentioned (1st paragraph).

2. Figure 2, spelling error, ''Ghan'' for ''Ghana''. Only 10 articles instead of 11 mentioned for the CRPA.

3. Please recheck the Table 1: Nogbou et al, study aim - grammatical error.

Olaitan et al, study aim - spelling error (carbapenemase-encoding genes)

Table 2: Abdeta et al, study aim - spelling error ( non-susceptible)

Codejoe et al, eligibility criteria for article selection includes only human isolates of

carbapenem resistant Acinetobacter baumannii (CRAB) and CRPA. This study, however,

reflects the bacterial isolates from the hospital environment. Please recheck for eligibility.

4. Please use a key for wording AST below tables 1 and 2.

5. Please realign the wording in tables 1 and 2 for similarity.

6. Table 1 and table 2, please check and recalculate the % prevalence for the CRAB and CRPA as many are incorrect,

e.g. table 1, CRAB prevalence for Musila et al is written as 56% despite all (27) isolates of Acinetobacter baumannii

being CRAB.

7. Table 2, Pseudomonas aeruginosa prevalence is missing in a few studies.

8. Figure 3 is not clear, very hazy.

9. The interpretation of the funnel plots are not described.

10. Please rewrite this sentence as it is ambiguous: ''Of the 14 articles analysed for the CRAB, (28.6%) were from South

Africa, Sudan and Uganda had 2(14.3%) each, while Ethiopia, Senegal Malawi, Kenya, Nigeria, and Sierra Leone had

one each''.

Discussion:

1. Please combine these sentences: ''Carbapenems are important broad-spectrum antimicrobials of last-resort, hence,

resistance to them signifies increased infection mortality, hospital stay duration, and cost of treatment (Friedman et al.,

2016). More importantly, when the infection is associated with notorious clinical pathogens such as A. baumanii and P.

aeruginosa.''

2.These sentences are redundant, please remove ''The true prevalence of these pathogens in sub-Saharan Africa is not well

known, especially the carbapenem-resistant strains. This study was carried out to determine the prevalence of the

carbapenem-resistant strains of A. baumanii and P. aeruginosa.''

3.This sentence does not make sense, please reword ''Thus, we believe that the current prevalence of antibiotic resistance

in A.baumannii infection is similar to the Sub-Saharan countries where these studies were carried out, most of which are

facing the same degree of severity of antibiotic resistance"'.

4. This sentence is confusing, advise rephrasing: A major strength of our study is that we included studies comprising of

non-disease-specific patients from patients, which ensured the representativeness of our estimates for these

institutions."

Conclusion:

1. Sentence very lengthy, please rewrite/reword: ''Urgent epidemiological studies through comprehensive

surveillance of the pathogen at both hospital and community-based that will showcase the

prevalence of CRAB in the environment, animal/animal products, and the hospital is advocated

to be conducted with the inclusion of sub-Saharan African countries currently lacking this data.''

References:

1. References appear inconsistent. Was a reference manager used?

2. Advise vancouver style of intext referencing preferred by the journal.

Reviewer #2: Dear editor,

Thank you for inviting me to review this manuscript that analyses AMR for these two important organisms.

Comments and suggestions

Abstract

Background:

Based on the latest publications carbapenems resistance rate were high in A. baumannii and dominant in P. aeruginosa, and respectively they would not be considered as the drugs of choice in either organism.

Introduction

In reference to Global burden of AMR, it would be advisable to mention attributable deaths to bacterial AMR.

Eligibility criteria, enhancement by recent publications.

There are some additional studies on ACIBA in the latest few months should be beneficial to mention, example: https://journals.plos.org/plosone/article?id=10.1371/journal.pone.0271355;

Results

Heading:

Prevalence of Carbapenem-resistant genes in A. baumanii and P. aeruginosa in sub-

Saharan Africa

Authors listed genes but no numbers or percentages from AST data reported, it would add value to the report.

It is interesting that authors haven’t quoted GLASS surveillance that have ACIBA aggregate data. It would add value to the discussion.

Please pay attention on typos.

END of COMMENTS

6. PLOS authors have the option to publish the peer review history of their article (what does this mean?). If published, this will include your full peer review and any attached files.

Reviewer #1: No

Reviewer #2: **Yes: **Olga Perovic

---

## [Author Response · Author response to Decision Letter 0]

25 Jan 2023

January 25, 2023

Adriano Gianmaria Duse, MD

Academic Editor

PLOS ONE

Dear Professor Adriano Gianmaria Duse,

Thank you for your reply regarding our manuscript PONE-D-22-27050 titled ‘Prevalence of carbapenem resistance in Acinetobacter baumannii and Pseudomonas aeruginosa in sub-Saharan Africa: a systematic review and meta-analysis.’

We are grateful for you and the reviewers’ comments, and the positive evaluation of our work. We have revised and modified the manuscript to meet PLOS ONE’s style requirements, including those for file naming. We have reconstructed the minor overlapping text with the publication stated in the PLOS ONE decision letter and cited all sources used in this article. 

Sincerely,

Margaret Toluwalayo Arowolo

Department of Microbiology

Faculty of Science

University of Lagos

Akoka-Yaba, Lagos

Nigeria

Journal Requirements: All the requirements stated have been revised and have been added to the manuscript.

- https://www.tandfonline.com/doi/full/10.1080/22221751.2019.1698273

-https://www.thefreelibrary.com/Prevalence+and+molecular+analysis+of+multidrug-resistant...-a0618470261

In your revision ensure you cite all your sources (including your own works), and quote or rephrase any duplicated text outside the methods section. Further consideration is dependent on these concerns being addressed

4. Please ensure that you refer to Figures 5, 6 and 7 in your text as, if accepted, production will need this reference to link the reader to the figure.

5. We note that Figure 2 in your submission contain map images which may be copyrighted. All PLOS content is published under the Creative Commons Attribution License (CC BY 4.0), which means that the manuscript, images, and Supporting Information files will be freely available online, and any third party is permitted to access, download, copy, distribute, and use these materials in any way, even commercially, with proper attribution. For these reasons, we cannot publish previously copyrighted maps or satellite images created using proprietary data, such as Google software (Google Maps, Street View, and Earth). For more information, see our copyright guidelines: http://journals.plos.org/plosone/s/licenses-and-copyright. The map was not copyrighted but was created in R using the rnaturalearth package

6. We note you have included a table to which you do not refer in the text of your manuscript. Please ensure that you refer to Table 1 and 2 in your text; if accepted, production will need this reference to link the reader to the Table.

Detailed response: we have addressed your editorial comments and responded to the comments by the reviewers as follows:

Response to Editor only (not for reviewers)

Reviewer #1: General comments:

1. Absence of page numbers and line numbers.- Page numbers and line numbers have been added to the revised manuscript.

2. Genus and species are not consistently italicised in the main text.- This has been corrected

3. Figures should not be included in the text according to journal recommendations.- This has been corrected

4. Spelling errors : ''Acinetobacter baumanii'' for Acinetobacter baumannii'', Discussion - ''tecnics'' for ''techniques'' table 1 and 2: ''disc'' instead of ''disk''.- This has been corrected

5. Word in full prior to using abbreviations: e.g. AST for antimicrobial susceptibility testing.- This has been effected

Abstract:

1. Extra comma after blaimp under results section.- The extra comma has been removed

2. Add the funnel plot analysis to the methods.- This has been added

3. This sentence is not mentioned/explained in the main text ''Carbapenem-resistant A. baumannii prevalence based on

sample source gave estimates of 24% (95% CI; 6 – 49; I2=99%; P<0.01). ''- This has been explained.

Introduction:- These points have been observed and corrected in the revised manuscript

1. It would be beneficial to mention the global statistics with regards to the prevalence of carbapenem resistant

Acinetobacter baumannii and Pseudomonas aeruginosa here.

2. Prevalence of the common carbapenemase genes worldwide amongst CRAB and CRPA should be highlighted here.

3. Please reword this sentence: ''This has made infection therapy more difficult and expensive, especially against the

notorious Gram-negative bacteria''

4. Define ''ESKAPE group.''

Methods: These points have been observed and corrected in the revised manuscript

1. Need to define the study period accurately and include the months in 2012 and 2022 as the start and end dates.

2. Were all studies that meet the eligibility criteria, irrespective of language, included?

Results: These points have been observed and corrected in the revised manuscript

1. For the carbapenem resistant Pseudomonas aeruginosa (CRPA), a total of 11 articles reviewed, however, only 9 articles

mentioned (1st paragraph).

2. Figure 2, spelling error, ''Ghan'' for ''Ghana''. Only 10 articles instead of 11 mentioned for the CRPA.

3. Please recheck the Table 1: Nogbou et al, study aim - grammatical error.

Olaitan et al, study aim - spelling error (carbapenemase-encoding genes)

Table 2: Abdeta et al, study aim - spelling error ( non-susceptible)

Codejoe et al, eligibility criteria for article selection includes only human isolates of

carbapenem resistant Acinetobacter baumannii (CRAB) and CRPA. This study, however,

reflects the bacterial isolates from the hospital environment. Please recheck for eligibility.

4. Please use a key for wording AST below tables 1 and 2.

5. Please realign the wording in tables 1 and 2 for similarity.

6. Table 1 and table 2, please check and recalculate the % prevalence for the CRAB and CRPA as many are incorrect,

e.g. table 1, CRAB prevalence for Musila et al is written as 56% despite all (27) isolates of Acinetobacter baumannii

being CRAB. – The % prevalence of CRAB and CRPA were calculated as a ratio of of the drug-resistant strains to the total sample size x 100.

7. Table 2, Pseudomonas aeruginosa prevalence is missing in a few studies.

8. Figure 3 is not clear, very hazy.

9. The interpretation of the funnel plots are not described.

10. Please rewrite this sentence as it is ambiguous: ''Of the 14 articles analysed for the CRAB, (28.6%) were from South

Africa, Sudan and Uganda had 2(14.3%) each, while Ethiopia, Senegal Malawi, Kenya, Nigeria, and Sierra Leone had

one each''.

Discussion: These points have been observed and corrected in the revised manuscript

1. Please combine these sentences: ''Carbapenems are important broad-spectrum antimicrobials of last-resort, hence,

resistance to them signifies increased infection mortality, hospital stay duration, and cost of treatment (Friedman et al.,

2016). More importantly, when the infection is associated with notorious clinical pathogens such as A. baumanii and P.

aeruginosa.''

2.These sentences are redundant, please remove ''The true prevalence of these pathogens in sub-Saharan Africa is not well

known, especially the carbapenem-resistant strains. This study was carried out to determine the prevalence of the

carbapenem-resistant strains of A. baumanii and P. aeruginosa.''

3.This sentence does not make sense, please reword ''Thus, we believe that the current prevalence of antibiotic resistance

in A.baumannii infection is similar to the Sub-Saharan countries where these studies were carried out, most of which are

facing the same degree of severity of antibiotic resistance"'.

4. This sentence is confusing, advise rephrasing: A major strength of our study is that we included studies comprising of

non-disease-specific patients from patients, which ensured the representativeness of our estimates for these

institutions."

Conclusion: The sentence has been corrected and reworded in the revised manuscript

1. Sentence very lengthy, please rewrite/reword: ''Urgent epidemiological studies through comprehensive

surveillance of the pathogen at both hospital and community-based that will showcase the

prevalence of CRAB in the environment, animal/animal products, and the hospital is advocated

to be conducted with the inclusion of sub-Saharan African countries currently lacking this data.''

References: The references have been corrected to meet the journal requirements

1. References appear inconsistent. Was a reference manager used?

2. Advise vancouver style of intext referencing preferred by the journal.- The references have been properly revised and corrections have been made. Vancouver style of intext referencing has been impleemented.

Reviewer #2: Dear editor,

Thank you for inviting me to review this manuscript that analyses AMR for these two important organisms.

Comments and suggestions

Abstract

Background:

Based on the latest publications carbapenems resistance rate were high in A. baumannii and dominant in P. aeruginosa, and respectively they would not be considered as the drugs of choice in either organism. – “drug of choice” has been changed to “effective drugs against bacterial pathogens”

Introduction

In reference to Global burden of AMR, it would be advisable to mention attributable deaths to bacterial AMR. – mentioned in lines 42-44

Eligibility criteria, enhancement by recent publications.

There are some additional studies on ACIBA in the latest few months should be beneficial to mention, example: https://journals.plos.org/plosone/article?id=10.1371/journal.pone.0271355;

Results : article did not meet the eligibility criteria

Heading:

Prevalence of Carbapenem-resistant genes in A. baumanii and P. aeruginosa in sub-

Saharan Africa – The article discussed both the prevalence of CRAB and CRPA as well as genes. We think the current title suits that.

---

## [Decision Letter · Decision Letter 1]

13 Jun 2023

Prevalence of carbapenem resistance in Acinetobacter baumannii and Pseudomonas aeruginosa in sub-Saharan Africa: a systematic review and meta-analysis

PONE-D-22-27050R1

Dear Dr. Arowolo,

We’re pleased to inform you that your manuscript has been judged scientifically suitable for publication and will be formally accepted for publication once it meets all outstanding technical requirements.

Kind regards,

Atef Oreiby, Ph. D.

Academic Editor

PLOS ONE

Additional Editor Comments (optional):

Please specify where to reach minimal data set in data availability statement.

**Comments to the Author**

1. If the authors have adequately addressed your comments raised in a previous round of review and you feel that this manuscript is now acceptable for publication, you may indicate that here to bypass the “Comments to the Author” section, enter your conflict of interest statement in the “Confidential to Editor” section, and submit your "Accept" recommendation.

Reviewer #1: All comments have been addressed

Reviewer #3: All comments have been addressed

2. Is the manuscript technically sound, and do the data support the conclusions?

Reviewer #1: Yes

Reviewer #3: Yes

3. Has the statistical analysis been performed appropriately and rigorously? 

Reviewer #1: Yes

Reviewer #3: I Don't Know

4. Have the authors made all data underlying the findings in their manuscript fully available?

Reviewer #1: Yes

Reviewer #3: Yes

5. Is the manuscript presented in an intelligible fashion and written in standard English?

Reviewer #1: Yes

Reviewer #3: Yes

6. Review Comments to the Author

Reviewer #1: Good day

There are minor errors noted.

line 75 - miss spelling of Klebsiella pneumoniae

line 76 - close bracket is missing

line 91 - change '' need to surveillance'' to '' need for surveillance''

line 177-179 - total analysis of CRPA does not equal 11, please recheck (this was mentioned previously)

page 11 - table, remove ''29'' from ''...teaching hospitals in Pretoria, South Africa''

page 19 - table, miss spelling of Doripenem under the carbapenem used column

Reviewer #3: The authors have responded adequately to comments raised in a previous round of review. However, there are some other minor corrections to do.

-Abstract section:

Lines, 48, 49 and 50: name of all genes should be italicized

-Introduction:

Line 61-62: Antimicrobial resistance and AMR: use only of the two words and rephrase the sentence to avoid repetition

Line 73: For ESKAPE group: Only cite the species of this group and remove all the other definition betwween brackets,

Line 85: delet the word (WASH)

-Discussion:

Line 261: replace A.baumannii by the species to ovoid repetition.

7. PLOS authors have the option to publish the peer review history of their article (what does this mean?). If published, this will include your full peer review and any attached files.

Reviewer #1: **Yes: **Prenika Jaglal

Reviewer #3: **Yes: **Larbi Zakaria NABTI

---

## [Editor Report · Acceptance letter]

22 Jun 2023

PONE-D-22-27050R1 

Prevalence of carbapenem resistance in Acinetobacter baumannii and Pseudomonas aeruginosa in sub-Saharan Africa: a systematic review and meta-analysis 

Dear Dr. Arowolo:

I'm pleased to inform you that your manuscript has been deemed suitable for publication in PLOS ONE. Congratulations! Your manuscript is now with our production department. 

Kind regards, 

on behalf of

Dr. Atef Oreiby 

Academic Editor

PLOS ONE